# Optomechanical synchronization across multi-octave frequency spans

Caique C. Rodrigues [1,2], Cauê M. Kersul [1,2], André G. Primo[1,2], Michal Lipson [3,4],
Thiago P. Mayer Alegre [1,2] & Gustavo S. Wiederhecker [1,2]✉

Experimental exploration of synchronization in scalable oscillator microsystems has unfolded a deeper understanding of networks, collective phenomena, and signal processing. Cavity optomechanical devices have played an important role in this scenario, with the perspective of bridging optical and radio frequencies through nonlinear classical and quantum synchronization concepts. In its simplest form, synchronization occurs when an oscillator is entrained by a signal with frequency nearby the oscillator's tone, and becomes increasingly challenging as their frequency detuning increases. Here, we experimentally demonstrate entrainment of a silicon-nitride optomechanical oscillator driven up to the fourth harmonic of its 32 MHz fundamental frequency. Exploring this effect, we also experimentally demonstrate a purely optomechanical RF frequency divider, where we performed frequency division up to a 4:1 ratio, i.e., from 128 MHz to 32 MHz. Further developments could harness these effects towards frequency synthesizers, phase-sensitive amplification and nonlinear sensing.

[1] Applied Physics Department, Gleb Wataghin Physics Institute, University of Campinas, Campinas, SP, Brazil. [2] Photonics Research Center, University of Campinas, Campinas, SP, Brazil. [3] Department of Electrical Engineering, Columbia University, New York, NY 10027, USA. [4] Department of Applied Physics and Applied Mathematics, Columbia University, New York, NY 10027, USA. ✉email: gsw@unicamp.br

Synchronization lies at the core of time keeping and underpins a vast class of natural phenomena, from life cycles to precision measurements[1]. In a nutshell, synchronization occurs when an oscillatory system has its bare frequency entrained by a weak external signal, which may have a slightly different tempo. Since its observation by Huygens in the 17th century, the synchronization of widely distinct systems has been shown to share remarkably universal features[1,2], fostering its exploration across many disciplines[3–5]. With the recent convergence among optical, mechanical and electrical waves using scalable microfabrication technologies, synchronization has emerged as a powerful tool targeted not only at technological applications, such as phase-lock loops in radio-based communications[6–8], but also at developing the fundamentals of chaotic systems[9], injection locking[10–12], electro and optomechanical devices[13–20], nonlinear dynamics[21–26], network coupling[27–30], and quantum synchronization[31–36].

Most synchronization realizations occur when the oscillation frequencies involved are barely dissimilar. This is usually the case because most oscillators rely on an underlying frequency-selective resonant response, e.g., mechanical, electrical, or optical resonance, which drastically suppresses off-resonant excitations. Despite the weak response to such nonresonant signals, oscillators with a strong nonlinearity may also synchronize when the ratio between external driving frequency ($\Omega_d$) and the oscillation frequency ($\Omega_0$) is close to a rational number $\rho = p/q$ called winding number[37], i.e., the ratio $\Omega_d/\Omega_0 = p/q$ with $p$, $q$ being coprime integers. Indeed, higher-order $p{:}q$ synchronization features have been experimentally observed in a variety of nonlinear systems, from Van der Pol's neon-bulb oscillator[38] to modern spin-torque oscillators[39–41], micro-electro-mechanical systems[42–47], delay-coupled lasers[9,48], nuclear magnetic resonance laser[49], and on-chip optical parametric oscillators[50]. These higher-order synchronization demonstrations are of major importance in radio-frequency (RF) division applications, which often demand low-power consumption and wide-band operation[51–53].

Within optomechanical devices, although seminal work has revealed that high-order synchronization is possible, its full strength is yet to be developed, potentially impacting the bridge between optical and RF signals[54] or enabling role in quantum[33,55,56] and classical devices[20,57]. For instance, the first optomechanical injection-locking demonstration by Hossein-Zadeh et al.[58] showed evidence of synchronization at $\Omega_d = 2\Omega_0$, while[59,60] demonstrated synchronization at subharmonics and the second harmonic in an on-fiber optomechanical cavity oscillator based on thermal effects. Theoretical work has suggested weak signatures of higher-order synchronization in optomechanical cavities[61].

Here, we experimentally demonstrate the entrainment of a silicon-nitride optomechanical oscillator (OMO) by an external signal up to two octaves away from its oscillation frequency. Furthermore, the OMO operates in the intriguing regime where higher-order synchronization ($p > q$) is actually stronger than the trivial 1:1 case, as determined by the degree of nonlinearity set by the laser frequency and intensity. Finally, we explore this regime to experimentally demonstrate a purely optomechanical radio-frequency divider with a phase noise performance better than the 1:1 locking regime. Our results open a route for exploring and engineering nonlinear synchronization in optomechanical oscillators[62], phase-sensitive amplification[63,64], nonlinear sensing[65], and collective dynamics of emerging oscillator arrays[30,66,67].

## Results and discussion

The general structure of optomechanical oscillators dynamic can be represented by the feedback diagram shown in Fig. 1a. The optical force driving the mechanical mode depends nonlinearly on the displacement, $x(t)$. Thus, the Lorentzian shape of the optical resonance provides a unique route to tailor the degree of nonlinearity of the optical force, defining how different harmonics of the mechanical oscillation are excited during the optical-to-mechanical transduction.

To establish synchronization, we apply a weak intensity modulation to the optical driving power, $P_{in}(t) = P_0[1 + \varepsilon \sin(\Omega_d t)]$, where $P_0$ is the continuous-wave average power and $\varepsilon$ ($\ll 1$) is the modulation depth. In the unresolved sideband regime, where $\Omega_0$ is smaller than the optical linewidth $\kappa$, the essence of the feedback loop of Fig. 1a is captured by introducing a delayed mechanical response $x(t) \rightarrow \tilde{x}(t - \tau)$, where $\tilde{x}$ is a normalized dimensionless displacement (details in the Supplementary Note 5). The optical force can then be efficiently written as a power series in $\tilde{x}(t - \tau)$,

$$F_{opt}(t) = f_{opt}\left[1 + \varepsilon \sin(\Omega_d t)\right] \sum_{n=0}^{\infty} F_n \tilde{x}^n(t - \tau), \qquad (1)$$

whose strength depends not only on the overall optical force strength, $f_{opt}$, but also on the dimensionless coefficients $F_n$, which dictates the intensity of the nonlinearity and their detuning dependence, as shown in Fig. 1b. Important optomechanical

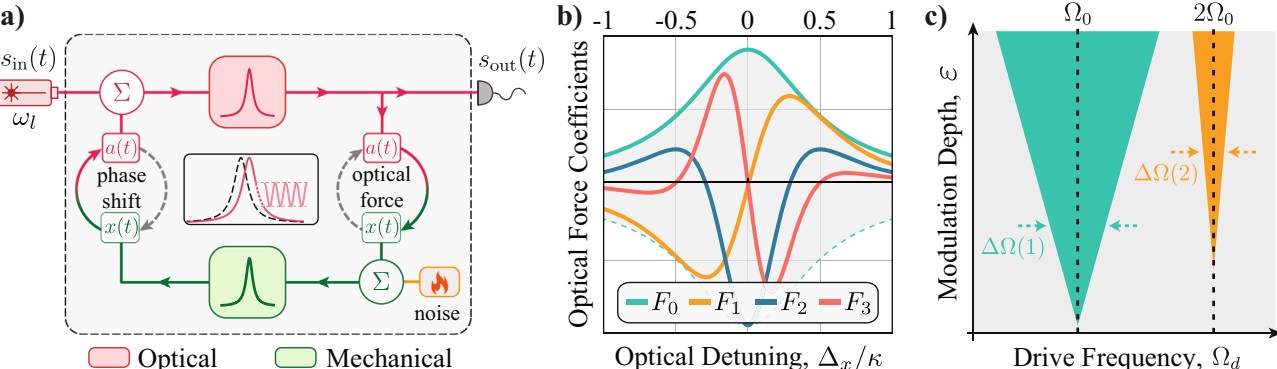

**Fig. 1 High-harmonic response of optomechanical oscillators. a** Optomechanical oscillator feedback diagram. The mechanical degree of freedom, $x(t)$, is initially in equilibrium with the thermal Brownian noise bath, but when a continuous-wave laser excites the optical field within the optical resonator, $a(t)$, the optical phase is imparted by the mechanical motion and transduced—via the optical resonance—to fluctuations on the optical energy. Due to radiation-pressure forces, the mechanical oscillator experiences a feedback (back-action) force that impacts its dynamics; **b** Optical force components as function of the optical detuning $\Delta_x = \omega_l - \omega_0 - Gx_0$ shown in Eq. (1) (details in the Supplementary Note 5); **c** Arnold tongues in the $\varepsilon - \Omega_d$ space illustrating 1:1 and 2:1 entrainment.

properties, such as optical cooling/amplification or spring effect[55,68], are described by considering up to the first-order term $F_1$ in Eq. (1). The modulation depth-dependent terms ($\propto \varepsilon$) enable the injection-locking and synchronization of the OMO to an external drive. While $F_0$ and $F_1$ hardly provide new insights into synchronization properties, the quadratic and cubic terms ($F_2$ and $F_3$) highlight a key aspect explored in this work: nonlinear synchronization properties can be adjusted with an easily accessible parameter, the optical detuning, which significantly changes their relative strengths, as shown in Fig. 1b.

The impact of these nonlinearities in the synchronization dynamics can be cast into the well-known Adler's model, which describes the slowly varying phase dynamics of an oscillator perturbed by a weak external drive[61,69]. Indeed, we show in "Methods" that the Taylor-series description of Eq. (1) leads to an effective Adler model when the optical modulation frequency is tuned towards a chosen harmonic of the mechanical frequency. Synchronization in this model arises when the perturbation strength overcomes the frequency mismatch between the drive and oscillator's harmonics. As the external drive frequency $\Omega_d$ is swept around the oscillator harmonics, the synchronization condition may still be satisfied and defines a region in a $\varepsilon - \Omega_d$ space known as Arnold tongues (ATs)[1], illustrated in Fig. 1c. Such response to higher harmonics could be readily explored for radio-frequency division, as we experimentally demonstrate for divisions ratio 2:1, 3:1 and 4:1, the same orders of the measured Arnold tongues maps.

To experimentally assess high-order synchronization and measure the ATs, it is important to harness the nonlinear response of an OMO. We achieve this control by employing a dual-disk optomechanical cavity based on silicon-nitride[70,71], as shown schematically in Fig. 2a. This cavity supports a relatively low frequency ($\Omega_m/2\pi = 31.86$ MHz) and high-quality factor mechanical mode ($Q_m = 1250$)[72], which is coupled to a transverse-electric optical mode ($Q_{opt} = 1.6 \times 10^5$ at a wavelength $\lambda \approx 1556$ nm) with an optomechanical coupling rate $g_0/2\pi = 16.2$ kHz. The experimental setup, shown in Fig. 2b, essentially consists of an intensity-modulated external cavity tunable laser that is coupled to the optomechanical cavity using a tapered fiber[70]. The output light is analyzed with an oscilloscope and an

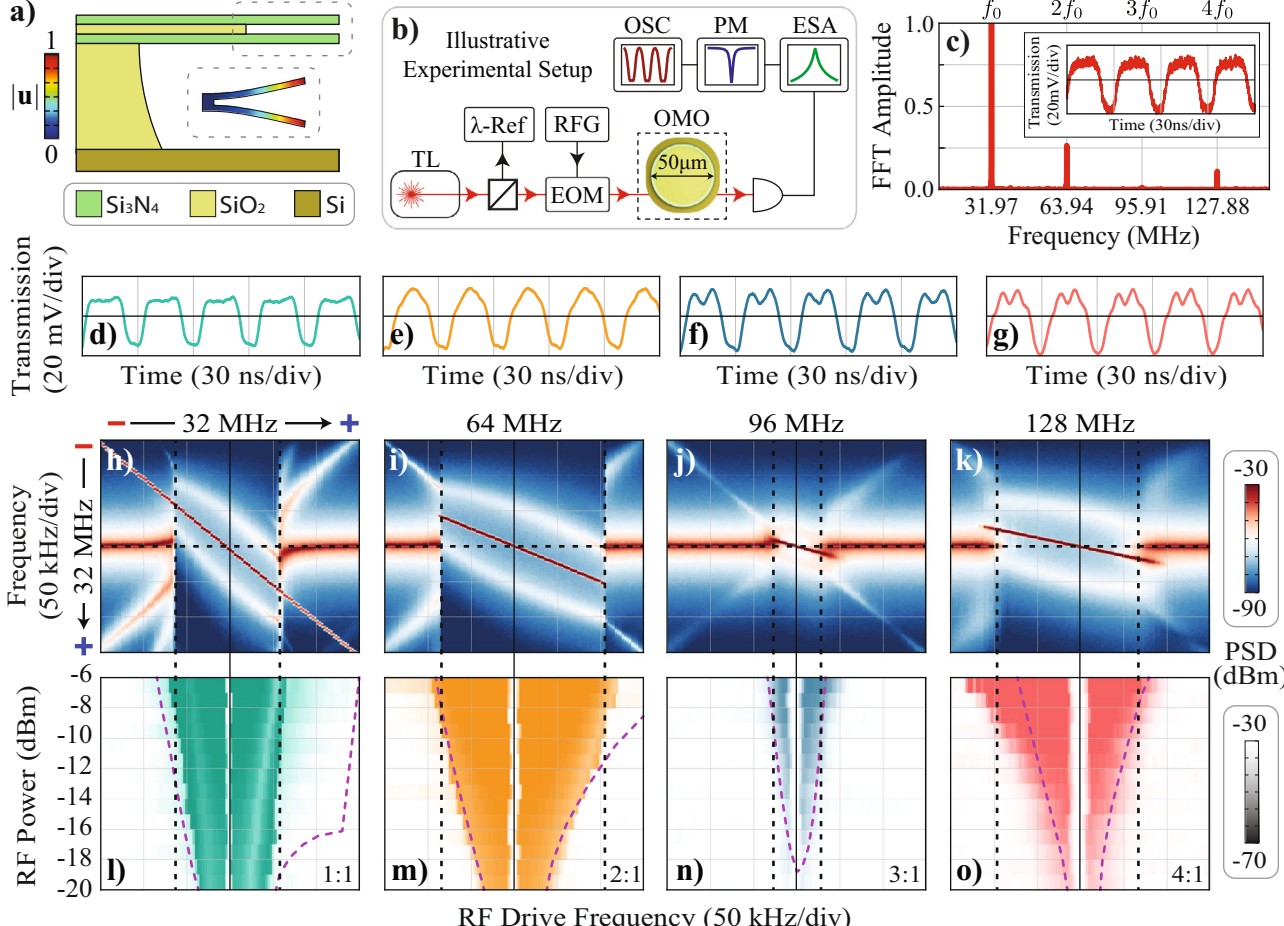

**Fig. 2 Experimental demonstration of multi-octave synchronization. a** Illustration of the silicon nitride dual-disk optomechanical cavity used in the experiment. The inset shows the simulated flapping mechanical mode displacement profile |**u**|; **b** Schematic of the experimental setup used; TL is the tunable laser source; λ-Ref: acetylene gas cell and a Mach-Zehnder interferometer as a reference in frequency; EOM: electro-optic modulator; RFG: radio-frequency generator; ESA: electrical spectrum analyzer; PM: power meter; OSC: oscilloscope; **c** Magnitude of the fast-Fourier transform of the OMO output signal (inset); **d–g** Time-trace of the OMO output entrained at $p = 1$ (**d**) until $p = 4$ (at **g**). A RF injection power of $-10$ dBm ($\varepsilon \approx 4\%$) was used; **h–k** RF spectrograms measured as the RF drive frequency sweeps from lower to higher frequencies around each OMO harmonic, $p = 1$ (**h**) until $p = 4$ (at **k**), for an injection RF power of $-10$ dBm. The vertical RF frequency axis is always centered at the mechanical oscillation frequency $\Omega_0/2\pi = 32$ MHz and increases from top to bottom, as the symbols minus and plus from **h**) suggests. The same is true for the horizontal axis, in which the RF drive frequency increases from the left to the right; **l–o** Measured Arnold tongues corresponding to each harmonic, obtained by stacking horizontal linecuts along the dashed black line shown in **h–k**. The purple curves are the simulated ATs and the color scale of each plot matches the grayscale range shown in the right.

electrical spectrum analyzer (ESA) that reveals the dynamics of the oscillator while monitoring the optical transmission.

To transition this optomechanical cavity into an OMO we raise the pump power to $P_0 = 480\,\mu W$ and fine-tune its wavelength such that the detuning between the laser frequency and the cavity resonance corresponds to $\Delta_x = 0.35\kappa$ ($\Delta_x/2\pi \approx 408$ MHz), which is inferred by monitoring the optical transmission. A typical OMO free-running output signal and the corresponding Fourier transform are shown in Fig. 2c, revealing the mildly nonlinear characteristic with a few noticeable harmonics. Interestingly, at this detuning, both the $F_0$ and $F_1$ terms in Eq. (1) are of similar strength (see Fig. 1b), suggesting that the nonlinear response to an injection signal should be readily observed. To observe injection-locking, the laser intensity modulation is activated, and the modulation frequency is swept around the OMO fundamental frequency or its harmonics ($p = 1-4$ and $q = 1$). The time-traces in Fig. 2d–g are captured with the injection signal frequency being precisely matched to each harmonic using a RF power of $-10$ dBm. As the RF driving frequency is detuned from each harmonic, the OMO response is monitored through the RF spectrum centered around the fundamental frequency $\Omega_0/2\pi$, as shown in the density plots of Fig. 2h–k. At the left-hand side of these plots, the RF tone is far away from the OMO harmonics and do not synchronize, thus, both oscillator and drive frequencies appear as distinct peaks, accompanied by nonlinear mixing products typical of driven oscillators[42]. When the RF tone approaches a harmonic, a clear transition occurs and a single RF peak emerges, which is one major signature of synchronization. The first striking feature is the observation of strong synchronization for all the driving harmonics, a phenomenon that has not been reported in optomechanical systems. Second, and most important, the width of the synchronization region for $p = 2$ and $p = 4$ are larger than the fundamental harmonic ($p = 1$). It is also remarkable that the $p = 3$ synchronization window is relatively small, counterposing the hierarchy among harmonics.

To map the synchronization window into Arnold tongues and understand the role played by the optical modulation depth, we performed the measurements shown in Fig. 2h–k for a range of RF powers, and built the ATs shown in Fig. 2l–o. The colored regions indicate a synchronized state and were obtained by stacking RF spectral slices along the OMO frequency, given by the horizontal dashed-lines in Fig. 2h–k. It is worth pointing out that the highest RF power ($-6$ dBm) corresponds to a modulation depth $\varepsilon \approx 6\%$, ensuring a weak perturbation regime. Although the existence of higher order tongues could be anticipated by qualitative analysis of the nonlinear terms in Eq. (1), further theoretical analysis is necessary to precisely picture their nature.

To study the observed AT behavior, we perform numerical simulations of the exact coupled equations describing both the mechanical and optical dynamics, and the resulting simulated Arnold tongues boundaries are shown in Fig. 3a. Despite the specific parameters that influence the precise behavior of the optomechanical limit cycles[61], such as optical detuning, optomechanical coupling, and optical/mechanical linewidths, a good agreement is observed between the measured and simulated tongues. Such agreement suggests that the observed features are indeed dominated by the optomechanical interaction itself, in contrast to silicon optomechanical devices where thermal and charge carrier effects strongly influence the self-sustaining oscillator dynamics[19,73]. Although the numerical model is useful for confirming the optomechanical nature of the observed effects, it hardly provides any analytical insight on the origins of the observed synchronization effects.

We obtain further insight by approximating the optical force as a delayed power series, as suggested in Eq. (1). This analysis

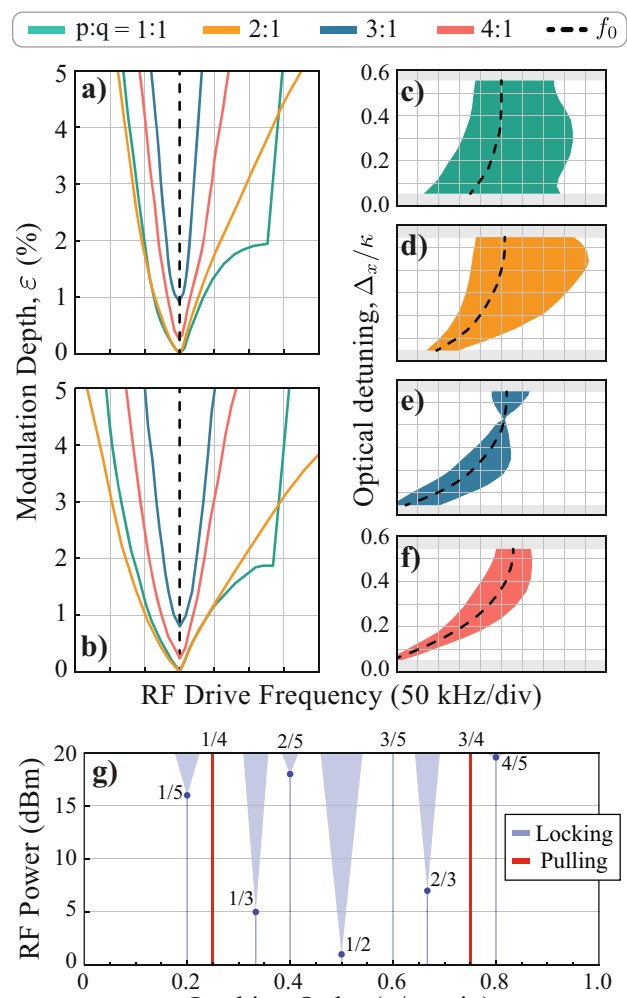

**Fig. 3 Numerical analysis and experimental observation of fractional synchronization. a** Arnold tongues boundaries simulated using the complete coupled optomechanical equations. The horizontal scale is the same used in experimental data of Fig. 2l–o, revealing a good agreement; **b** Same simulation done at (**a**) but now considering only one parametric term in each simulation, i.e., the green (1:1) boundary was simulated considering $\varepsilon F_1 = \varepsilon F_2 = \varepsilon F_3 = 0$ but $\varepsilon F_0 \neq 0$ (details in the Supplementary Note 5). The orange (2:1) boundary has only the term $\varepsilon F_1 \neq 0$, the blue (3:1) has $\varepsilon F_2 \neq 0$ and the red (4:1) has $\varepsilon F_3 \neq 0$; **c-f** Impact of optical detuning $\Delta_x$ in the ATs, showing their tunability and the possibility of a vanishing $p = 3$ tongue at $\Delta_x \approx 0.43\kappa$ for the parameters used. These maps were simulated using $\varepsilon = 5\%$ and the black-dashed line is the mechanical oscillation frequency $f_0$, which increases with $\Delta_x$ because of the optical spring effect; **g** Measured fractional synchronization threshold, indicated as blue dots, to observe a finite-width AT. The red lines indicate the locking orders that did not synchronize and only frequency pulling was observed. The Arnold tongues shown are illustrations (see Supplementary Note 2 for actual data).

allows exploring the synchronization role of each nonlinear component $F_n$ in Eq. (1) and elucidates the underlying structure of high-harmonic synchronization. The nonlinear components that *are not* proportional to the driving signal define a "forced Van der Pol-Duffing oscillator" responsible for the oscillator limit cycle observed in Fig. 2c.

The synchronization dynamics is related to the terms proportional to the RF driving signal ($\propto \varepsilon$). However, in addition to the usual non-parametric excitation ($\propto \varepsilon F_0$), the injection signal also contributes to time-dependent coefficients in the mechanical

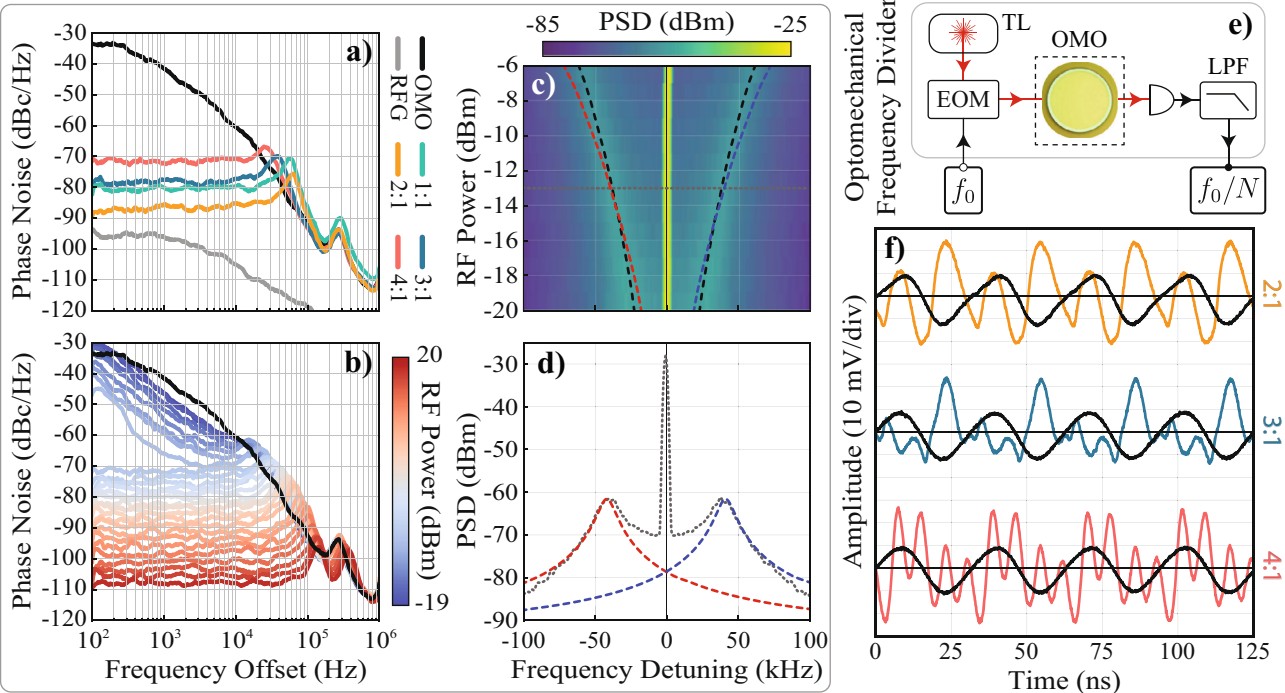

**Fig. 4 Phase-noise reduction and optomechanical frequency division. a** Measured one-sided phase noise spectral density for the free-running (black), injection-locked (colored), and RF injection signal (gray). The RF power used for the injections was −7 dBm ($\varepsilon \approx 5.5\%$); **b** Phase noise spectral density evolution as a function of the RF power for the 4:1 injection; **c** Comparison between experimental data sidebands and the semi-analytical model prediction (see Supplementary Note 7). The colormap is the experimental power spectral density (PSD) around the OMO fundamental frequency. The RF drive frequency was set to the OMO frequency for all the RF powers shown. The black dashed lines are the experimental fit and the red/blue curves are the semi-analytical prediction, showing excellent agreement; **d** Experimental PSD, in gray, for a RF power of −13 dBm ($\varepsilon \approx 2.75\%$), which is the same horizontal dashed gray linecut at (**c**) showing the agreement between semi-analytical model and experimental data for these sidebands, for both frequency and linewidth; **e** Schematic of the optomechanical frequency divider; LPF means "low-pass filter"; **f** Experimental Optomechanical frequency division. The orange, blue and red curves are the injection-locked signal output from the OMO for the cases 2:1, 3:1, and 4:1, respectively. The overlapping black curves are the divided signal obtained using a low-pass filter with 48 MHz cutoff frequency.

oscillator dynamical equation. Physically, these time-varying coefficients indicate that the external signal modulates the oscillator's frequency and damping properties, leading to linear ($\propto \varepsilon F_1$) and nonlinear ($\propto \varepsilon F_{2,3}$) parametric resonance effects, a situation resembling the dynamics of a nonlinear Mathieu equation[71,74].

By neglecting all but one time-dependent term in the numerical simulations, we could identify how each harmonic ($p = 1\text{--}4$) is related to the force expansion coefficients shown in Fig. 1b. The resulting map is shown in Fig. 3b, where each boundary was simulated considering only one parametric term, while all the others were set to zero. The resemblance with the full model simulation at Fig. 3a is remarkable. This analysis reveals that each $\varepsilon F_{p-1}$ term in the force expansion is the leading contribution to the $p$:1 AT, for all measured harmonics. For instance, as the $p = 3$ entrainment occurs due to the $\varepsilon F_2$ parametric term, the thinner tongue observed in Fig. 2n is explained by the negligible value for $F_2$ at this detuning. Interestingly, although quadratic force terms like $F_2 x^2$ are often ignored in nonlinear mechanical oscillators (as they arise from an asymmetric elastic potential energy), here, they emerge naturally from the Lorentzian shape of the optical mode and can be tuned with the optical detuning. Another interesting feature, present both in the analytical and numerical model, is the presence of a cusp in the 1:1 tongue at −16 dBm RF power. Although we verified that such a feature occurs due to an amplitude bifurcation using the analytical model (see Supplementary Note 8), the cusp was not observed in the experimental trace.

The insights brought by our semi-analytical model suggest that tunable Arnold tongues should be feasible. In Fig. 3c–f we show a full numerical simulation of the ATs as a function the optical detuning, confirming this possibility. In particular, a complete suppression of $p = 3$ tongue is attainable (Fig. 3e). Such rich response to higher harmonic excitation led us to verify whether our OMO could also respond to fractional frequency excitation, i.e., where $p/q$ is not an integer number. These experimental results are summarized in Fig. 3g but the full map can be found in Supplementary Note 2 for various subharmonics of the mechanical frequency, revealing terms of the famous Farey sequence known in number theory[37]. Note, however, that the injection signal power required to observe fractional tongues was substantially larger, with some fractions (e.g., 4/5) requiring a full modulation, which is beyond the reach of our semi-analytical approximations ($\varepsilon \approx 100\%$).

**Optomechanical frequency division.** An important aspect often praised when investigating synchronization and injection-locking phenomena is the reduction of phase noise (PN) in free-running oscillators. While optomechanical oscillator's phase noise (PN) has been previously explored[15,58,67,73,75], its characteristics under high harmonic injection are not known. In Fig. 4a we show the measured PN at the fundamental oscillator's frequency for the free-running OMO and injection-locked at the harmonics $p = 1\text{--}4$ (see "Methods" for details). The PN curves were taken using a constant RF power of −7 dBm ($\varepsilon \approx 5.5\%$) for all harmonics. The general behavior of the free-running OMO PN has been

discussed previously[75] and it is influenced by various noise sources, such as flicker, thermomechanical, and amplitude-to-phase conversion[76]. When injection locked at $p = 1$ (green curve), the PN performance improves significantly, and the PN of the higher harmonics are surprisingly low, despite that the same modulation depth was employed. Indeed, the $p = 2$ injection offers an improvement over the trivial $p = 1$ case, $p = 3$ is slightly deteriorated, and $p = 4$ PN suffers a significant penalty of 10 dBc/Hz at small offset frequencies, however, it still preserves the low-frequency PN plateau, characteristic of injection-locked oscillators. To investigate the RF power dependence of each harmonic, PN curves were measured over a range of RF powers, shown in Fig. 4b for the 4:1 case. The transition to a low-frequency PN plateau (around $-7$ dBm) observed in Fig. 4b also occurs for other harmonics, albeit at lower injection powers, showing that very low PN levels can be achieved at the expense of higher RF power levels (see Supplementary Note 3 for other harmonics). In particular, while the 1:1 PN of Fig. 4a reaches $-80$ dBc/Hz at $-7$ dBm, the 4:1 PN of Fig. 4b requires $-1$ dBm to reach $-80$ dBc/Hz, still corresponding to a moderate modulation depth of 11%. A qualitative understanding of the observed PN behavior can be cast upon previous investigations in the context of superharmonic injection-locking[7,77–79]. When the injection-signal PN is negligible, the phase-noise of a superharmonic injected oscillator is written as

$$\mathcal{L}_{\text{out}}(\Omega) = \frac{\mathcal{L}_{\text{free}}(\Omega)}{1 + (\Delta\Omega_n/\Omega)^2 \cos^2\theta}, \tag{2}$$

where $\mathcal{L}_{\text{free}}(\Omega)$ is the free-running OMO PN spectra, i.e., the black curve of Fig. 4a; $\Delta\Omega_n$ is the locking range (AT width) for each harmonic; $\theta$ is the phase offset between the injection signal and the OMO. Apart from the phase offset $\theta$, the AT width determines the locking range and is often associated with good phase noise performance.

Indeed, the wider lock range $\Delta\Omega_2$ observed for the 2:1 injection is associated with a better PN. For the 3:1 and 4:1 PNs cases, however, the trend is not as clear. While the phase-noise is reduced as the lock-range increases (due to higher injection power), the 4:1 PN shown in Fig. 4a is not lower than the 3:1 injection, despite the wider 4:1 tongue. Although it is not clear all the factors contributing to this discrepancy, we verified in numerical simulations that the phase-offset $\theta$ varies among harmonics and could partially contribute to the observed mismatch. One unique factor contributing to these phase offsets in nonlinear oscillators is the strong frequency pulling[80,81] that distinctively shifts the bare OMO frequency for each harmonic. Indeed, we can notice in the injection maps of Fig. 2h–k that the locking frequency loci are not symmetric relative to the OMO frequency. For example, Fig. 2o is shifted towards lower frequencies, while Fig. 2m shifts toward higher frequencies. Such shifts are also anticipated by our semi-analytical model and can be traced back to the effective perturbation strength and frequency mismatch in Adler's model (see "Methods"). These nonlinearities also highlight the weakness of neglecting the amplitude-phase coupling in the PN modeling of OMOs.

Another feature that supports the amplitude-phase coupling effects in the PN spectrum, which is not readily captured by the simple model leading to Eq. (2), is the presence of the sidebands appearing in Fig. 4a between 20 and 60 kHz. In contrast to the fixed-frequency satellite peaks at 150 kHz, which are caused by parametric mixing with a spurious mechanical mode, these peaks are intrinsic to the nonlinear locking dynamics of OMOs. These sidebands were discussed by Bagheri et al.[20] and attributed to the coupling between phase and amplitude dynamics that are intrinsic to OMOs. Based upon our amplitude-phase model leading to the effective Adler equations (Eq. (4)), we derive a quantitative model, in similarity to spin-torque oscillators[40], which predicts both frequency splitting and linewidth of these sidebands. Despite the various approximations necessary, the fitted model agrees remarkably well with the experimental data, as shown in blue/red curves in Fig. 4c, d.

In the context of higher-order synchronization, the demonstrated phase-noise performance could be explored towards injection-locked superharmonic frequency dividers[7,8], which generate radio-frequency signals at a fraction of a higher frequency reference. Despite the low power-consumption advantage of injection-locked dividers, compared to other technologies, such as regenerative and parametric dividers[8], they often suffer from a narrow lock range. While OMOs offer intrinsically narrower lock ranges compared to electronic injection dividers[8], the wide Arnold tongues reported in Fig. 3 suggest that a robust OMO frequency division is feasible. Exploring this strong response to higher harmonics, the experimental schematic of Fig. 4e was implemented to perform the demonstration of an optomechanical frequency division. A low-pass RF filter (48 MHz cutoff, MiniCircuits SLP-50+) rejects the higher-harmonics generated by the injection-locked OMO and delivers an output signal at a fraction of the injected reference, $f_0/N$.

The measured frequency-divided signals for 2:1, 3:1, and 4:1 locking for a RF power of 0 dBm are shown in Fig. 4f. The worst PN performance, obtained in the divide-by-4 case, is better than $-70$ dBc/Hz and can be significantly improved at higher RF powers, as shown in the red-tone traces in Fig. 4b. Further improvement in phase-noise could be achieved by using devices with higher mechanical quality factor and stronger optical driving power, for instance, double-disk optomechanical devices with mechanical quality factors exceeding $10^4$ and driven at larger amplitudes (using higher optical power) could exhibit a further PN reduction of 30 dB (see Supplementary Note 3). These results show that OMO-based frequency dividers can be readily derived from the observed higher-order synchronization. Although there is room for improvement in optomechanical frequency dividers, its ability to generate frequency references in the optical domain could be explored in experiments requiring optical synchronization, such as radio antenna telescopes[82], optical frequency combs[50], or coherently linking arrays of optomechanical oscillators with distinct frequencies[67]. Given the current state-of-the-art in hybrid integration[83] and electro-optical conversion in photonic circuits[73], the demonstrated divider could still ensure the low power consumption expected for injection locking frequency division.

We have experimentally demonstrated an optomechanical oscillator entrained by high-order harmonics and its application as a purely optomechanical frequency division. The wider locking range observed for the higher harmonics, and its theoretical mapping to each nonlinear term in the oscillator dynamics, open new routes to control nonlinear synchronization phenomena in optomechanical oscillators, including the tailoring of the nonlinear response through the laser-cavity detuning and frequency synthesizers optomechanical devices. Furthermore, the importance of nonlinear parametric effects could also significantly impact phase-sensitive amplification[84] and nonlinear sensing[65] with optomechanical devices. The demonstrated entrainment should also enable novel configurations for coupling and controlling optomechanical arrays based on dissimilar resonators. The demonstration of locking at fractional harmonics could also be a starting point for further nonlinear dynamics investigations within an optomechanical platform.

## Methods

**Optical energy**. The optical energy dependence on the laser-cavity detuning and mechanical displacement is given by,

$$|a|^2 = \frac{\kappa_e}{(\Delta - Gx)^2 + \kappa^2/4} P_{\text{in}}, \qquad (3)$$

in which two key parameters that will enable the tuning of the OMO nonlinear response arise: the input laser power, $P_{\text{in}}$, and the bare optical detuning, $\Delta = \omega_l - \omega_0$, between the pump laser ($\omega_l$) and optical mode ($\omega_0$) frequencies; $x$ is the mechanical mode amplitude, $G = \partial\omega/\partial x$ is the optomechanical pulling-parameter, $\kappa$ is the optical mode linewidth and $\kappa_e$ is the external coupling to the bus waveguide[3].

**Effective Adler model**. By employing the Krylov–Bogoliubov–Mitropolsky time-averaging method[85] at the mechanical oscillator equation, an effective Adler's equation may be derived (details in the Supplementary Note 6),

$$\dot{\Phi} = \nu(\rho) + \varepsilon\frac{\Delta\Omega(\rho)}{2}\sin(\rho\Phi). \qquad (4)$$

where $\Phi$ is the mechanical oscillator phase correction and $\dot{\Phi}$ denote its time derivative; $\nu(\rho)$ is the mean correction of $\Omega_0$ and $\Delta\Omega(\rho)$ is the size of the synchronization window at a particular harmonic $\rho = p/q$. Although many approximations must be carried on, this analysis relates the Taylor series coefficients in Eq. (1) with the coefficients $\nu(\rho)$ and $\Delta\Omega(\rho)$ in the effective Adler's model Eq. (4), providing a quantitative description of the width hierarchy among the measured ATs.

**Experimental setup**. A full schematic of the experimental setup is shown in Supplementary Note 1, along with optical and mechanical characterization of the bare resonator data. The optical transmission and the RF spectral measurements for the bare resonator properties were taken at low pump powers (<50 μW). The laser wavelength and detuning are accurately monitored using a Mach-Zehnder Interferometer and a HCN gas cell. The cavity is inside a vacuum chamber with pressure of approximately 0.1 mbar and at room temperature. Finally, the transduced signal goes to two detectors: a power meter (PM) that will track the optical mode and a fast photodetector (NewFocus 1617AC Balanced Photodetector) with 800-MHz bandwidth whose electrical output feeds both the electric-spectrum analyzer (ESA, Keysight N9030A) and oscilloscope (OSC, DSO9254A). The phase-noise measurements were performed in the spectral domain using the ESA N9030A phase-noise measurement application (N9068A). There was also a feedback loop between the PM and the TL to lock the signal, preventing the optical resonance to drift due to unwanted external perturbations.

**Phase noise**. To derive the approximate expression for the phase noise (Eq. (2)), we must start from the general PN expression[7,77],

$$\mathcal{L}_{\text{out}}(\Omega) = \frac{(\Delta\Omega_n/n)^2 \mathcal{L}_{\text{inj}}(\Omega)\cos^2\theta + \Omega^2 \mathcal{L}_{\text{free}}(\Omega)}{\Delta\Omega_n^2\cos^2\theta + \Omega^2}. \qquad (5)$$

Since the injection-locking signal is derived from a stable RF frequency source (Agilent PSG E8251), $\mathcal{L}_{\text{inj}}(\Omega)$, the injection signal PN spectra are orders of magnitude smaller than $\mathcal{L}_{\text{free}}(\Omega)$, and then $\mathcal{L}_{\text{inj}}(\Omega)/\mathcal{L}_{\text{free}}(\Omega) \to 0$ results in Eq. (2). The modulation depth as a function of the RF power is given by $\varepsilon = \pi\sqrt{P_{\text{RF}}R}/V_\pi$, where $R = 50\,\Omega$ and $V_\pi = 5.5$ V is the optical modulator parameter. The phase angle is given by $\theta = \arcsin\left[(\Omega_0 - \Omega_d/n)/\Delta\Omega_n\right]$. A more detailed analysis is given in Supplementary Note 3 where we show the measured phase noise as a function of the RF power for all the harmonics.

**Simulations**. The acquired data were compared with numerical simulations using Julia language together with well-known and powerful packages like Differential Equations.jl, DSP.jl and Sundials.jl. As we are dealing with a stiff system, i.e., there is more than one relevant natural time scale that differ by many orders of magnitude, solvers available in Julia offer a better performance. We simulate the system for a range of modulation depths $\varepsilon$ while the RF signal sweeps around a set of chosen $p{:}q$ region, revealing the nature of synchronization. With the obtained time trace, we then locally Fourier transformed the data to construct the spectrogram. A detailed discussion on the numerical simulation is available at Supplementary Note 4. The mechanical mode effective mass and the zero point fluctuation were obtained from COMSOL Multiphysics finite element simulations, $m_{\text{eff}} = 101.82$ pg, $x_{\text{zpf}} = 1.536$ fm, leading to an optomechanical pulling parameters $G/2\pi = (g_0/2\pi)/x_{\text{zpf}} = 10.546$ GHz/nm.

## Data availability

Further data supporting the findings of this study are openly available at Zenodo at https://doi.org/10.5281/zenodo.4737381.

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

## Acknowledgements

This work was supported by São Paulo Research Foundation (FAPESP) through grants 2019/14377-5, 2018/15577-5, 2018/15580-6, 2018/25339-4, 2017/24845-0, 2020/06348-2, 2019/09738-9, Coordenação de Aperfeiçoamento de Pessoal de Nível Superior—Brasil (CAPES) (Financial Code 001). This work was performed in part at the Cornell NanoScale Science and Technology Facility, which is supported by the NSF, its users, and Cornell University.

## Author contributions

C.C.R. and G.S.W. designed the experiment; C.C.R. performed measurements and data analysis with help from C.M.K. and A.G.P.; C.C.R., C.M.K., and A.G.P. contributed to the theoretical framework. G.S.W. and M.L. designed and fabricated the device; T.P.M.A. and G.S.W supervised the project. All authors contributed to the discussions and pre-paration of the manuscript.

## Competing interests

The authors declare no competing interests.
