## [Peer Review File · Nature Communications]

REVIEWER COMMENTS

Reviewer #1 (Remarks to the Author):

In this manuscript, the authors fill a gap in the synchronization of optomechanical systems by experimentally demonstrating the injection locking of an oscillator (frequency Ω_0) to an external driving frequency Ω_d . Here, $\Omega_d/\Omega_0=p/q$ is a rational number.

In Fig. 2 they show Arnold tongues (synchronization regions) for the case of a harmonic drive frequency (p/q ranges from 1 to 4). These data are complemented by experimental results demonstrating locking to subharmonic frequencies (p/q ranges from 1/5 to 4/5). The simulation is shown in Fig. 3(d), the experimental data in Fig. 3 of the Supplementary Information. The subharmonic case requires substantially larger signal strengths, however.

Finally, the authors demonstrate the reduction of phase noise for injection locking with p/q ranging from 1 to 4.

These are interesting and original experimental results that are analyzed in an appropriate way. In my opinion, the manuscript is suitable for publication in Nature Communications once the following points have been addressed:

p. 1 right column:

"Here, we experimentally demonstrate the entrainment...by an external signal up to four octaves away from its oscillations frequency".

one octave = factor 2 in frequency

four octaves = factor 16 in frequency.

So shouldn't the statement be "...up to two octaves away..."

Same for the phrase "...several octaves away..." in the abstract.

Fig. 2 What is the meaning of the cusp in the purple simulation curve in panel (d3)

Ref. [1] is a book review of Pikovsky et al.'s textbook.

I assume [1] is meant to be a reference to the textbook itself?

There are technical problems with many of the references, e.g., missing page numbers in [8], [15], [20], incorrect page numbers "1", etc.

The manuscript needs editing in many places.

See, e.g., the first phrase of the introduction

(where "phenomena" should be omitted, or "lies" and "underpins" be replaced by "lie" and "underpin"),

or Ref. [5] of the Supplementary Information.

Reviewer #2 (Remarks to the Author):

This manuscript studies synchronization of the mechanical oscillation of an optomechanical system to a radiation pressure drive. Synchronization is an important and rich phenomenon, observed in many different contexts and with significant applications.

The paper is solid, including some very clean and elegant result together with a careful comparison of simulations to experiments. I find the synchronization data in Fig. 2 d2-g2) particularly beautiful. I commend the authors on a very nice set of experiments.

This said, I'm not sure that the work is suitable for Nature Communications. My most major concern is the level of advance reported in the manuscript. The physical technology (a silicon nitride double disk resonator) is well established, and the key results - that synchronization is possible by driving at harmonic and fractional frequencies - are extensions of prior work, rather than entirely new demonstrations. In particular, as acknowledged by the authors their Ref. [57] shows synchronization to a second harmonic drive. Less explicitly acknowledged by the authors is that their Ref. [58] shows synchronization to a fractional drive. I would note that Physical Review E 91 032910 (2015) also demonstrates second harmonic synchronization.

The present work goes further than these previous works, demonstrating synchronization up to the fourth harmonic, and showing explicitly through data processing that it is possible to do frequency division using harmonic synchronization. It also includes the interesting observation that the synchronization bandwidth is larger for the second and fourth harmonic than for the fundamental, and that this has implication for the phase noise of the synchronized oscillator. Nevertheless, to me these all feel like solid progress, but not the dramatic leap one might expect for Nature Communications.

In my view, the most novel aspect of the publication is the explicit demonstration of frequency division (even if only through data processing). However, here I feel that the paper is lacking the rigor that is found elsewhere in it. Perhaps the most important unanswered question is the relative quality (in terms of phase noise) of the divided signal compared to the input radio frequency signal. I would expect that this signal will be of far inferior quality, raising the question of the true usefulness of this form of optomechanical frequency division. Were the authors to do a direct comparison and show rigorously that the divided phase noise was comparable to that on the input radio frequency signal, or show through simulations/theory that there was a believable path to get there, this would significantly increase the novelty of the paper in my mind.

Some other points:

- 1) Paragraph 3 of the introduction is not very clear. The first sentence says that higher-order synchronization has been "overlooked", Then "For instance..." lists examples where it has not been overlooked, rather than evidence that it has been overlooked as implied by the "For instance". As discussed above, there is also at least one reference missing here, as well as an acknowledgement that Ref. [58] showed fractional synchronization.
- 2) It isn't exactly clear to me what the vertical axis label "AT Tip (dBm)" means in Fig. 3d. I would define this explicitly in the caption.
- 3) The 4:1 phase noise curve in Fig. 4a seems qualitatively different than the other curves. Can the authors comment on this?
- 4) The last paragraph on Page 6 comparing optomechanical and other frequency dividers would benefit from a clearer discussion of pros and cons. E.g. Why do OMO type dividers potentially offer lower power-consumption? ...and the authors point out a limitation of OMOs (I think, the wording isn't clear), that they have limited locking range. They then don't discuss how their approach, presumably, helps overcome this?

Summary of response

We thank all referees for their careful reading of our manuscript. To fully address all the concerns raised, we performed new experiments, and added previously acquired experimental data to the revised manuscript. Most of these changes were incorporated directly into the text and figures, which can be more easily checked with the accompanying marked-up version. When necessary, we also included additional data and discussion in the supplementary information. Below we summarize the main changes in the manuscript. In order to ease the evaluation of this new version, we provide a marked up version of the manuscript.

Note: All line numbers mentioned in the response letter refer to the line-numbered MARKED UP version of the manuscript.

1. Main text

- a. We performed new experiments to demonstrate (without resorting to data processing) optomechanical frequency division for the 2:1, 3:1, and 4:1 injection locking.
- b. A new dataset (previously acquired) is also presented showing the Phase noise spectra evolution with injection signal power to demonstrate the potential of achieving low phase noise using higher order injection locking.
- c. Following this new data acquisition, we provide an updated Fig. 4 which we believe shall address important concerns from Referee 2.
- d. As suggested by Referee 2, we wrote a paragraph on how to further improve frequency division phase noise performance.
- e. We improve the explanation of Fig 3d, as pointed out by Referee 2;
- f. We corrected the abstract and introduction when quoting "several octaves", following suggestions from Referee 1.
- g. Following Referee 1 advice, we briefly discussed the Cusp behavior and added extended simulation data showing its amplitude-bifurcation origin.

2. Supplementary Material

- a. We wrote a new section (Section III) about phase noise details, including extra experimental data which was not shown in the main text;
- b. A section dedicated to discuss the cusp behavior in the 1:1 Arnold tongue , as pointed by referee 1.
- c. We made several minor adjustments in the supplementary text and presentation.

3. Other changes

- a. Several English-typoes were adjusted throughout the manuscript;
- b. All the references were revisited and fixed, as well as a few new references were added.
- c. In order to address Referee 2 concerns, we modified Figure 4 and moved the old Figure 4(b) to Supplementary Information section III

Referee 1

Note: All line numbers mentioned in the response letter refer to the line-numbered MARKED UP version of the manuscript.

In this manuscript, the authors fill a gap in the synchronization of optomechanical systems by experimentally demonstrating the injection locking of an oscillator (frequency Ω_0) to an external driving frequency Ω_d . Here, $\Omega_d/\Omega_0=p/q$ is a rational number.

In Fig. 2 they show Arnold tongues (synchronization regions) for the case of a harmonic drive frequency (p/q ranges from 1 to 4). These data are complemented by experimental results demonstrating locking to subharmonic frequencies (p/q ranges from 1/5 to 4/5). The simulation is shown in Fig. 3(d), the experimental data in Fig. 3 of the Supplementary Information. The subharmonic case requires substantially larger signal strengths, however.

Finally, the authors demonstrate the reduction of phase noise for injection locking with p/q ranging from 1 to 4.

These are interesting and original experimental results that are analyzed in an appropriate way. In my opinion, the manuscript is suitable for publication in Nature Communications once the following points have been addressed:

R: We thank the referee for this careful review and point out that we performed new experiments to emphasize the novel demonstration of RF frequency division without resorting to a numerical low pass filter (see updated Fig. 4). Together with the new phase noise data, also in Fig. 4, we believe that the potential of optomechanical frequency division is clearer in the revised manuscript.

R1-Q1) p. 1 right column:

"Here, we experimentally demonstrate the entrainment...by an external signal up to four octaves away from its oscillations frequency".

one octave = factor 2 in frequency

four octaves = factor 16 in frequency.

So shouldn't the statement be "...up to two octaves away..."

R: We agree with the referee. This mistake was corrected in the revised manuscript. The text between lines 60-65 was adjusted.

R1-Q2) Same for the phrase "...several octaves away..." in the abstract.

R: This was also corrected in the abstract.

R1-Q3) Fig. 2 What is the meaning of the cusp in the purple simulation curve in panel (d3)

R: The cusp in the simulated tongues originates from a bifurcation in the oscillator's amplitude when the modulation depth is increased. It is not clear, however, why we could not observe the cusp in our experimental data. We learned from simulations that the Arnold tongues do exhibit hysteresis when the RF frequency sweep direction is reversed, this could be one route in future experimental investigation. In order to clarify its origin, we pointed the reader (around line 260) to a new Sec. V.C in the supplementary material that shows good

agreement between the analytical prediction of the amplitude bifurcation points with the cusp observed in the full numerical solution.

R1-Q3) Ref. [1] is a book review of Pikovsky et al.'s textbook. I assume [1] is meant to be a reference to the textbook itself? There are technical problems with many of the references, e.g., missing page numbers in [8], [15], [20], incorrect page numbers "1", etc.

R: Indeed there were several mistakes in the reference list, we carefully checked and corrected each reference in the revised manuscript.

R1-Q4) The manuscript needs editing in many places. See, e.g., the first phrase of the introduction (where "phenomena" should be omitted, or "lies" and "underpins" be replaced by "lie" and "underpin"), or Ref. [5] of the Supplementary Information.

R: We performed a thorough english grammar revision to correct the highlighted mistakes and others we identified throughout the text. These can be tracked in the accompanying marked up manuscript.

Referee 2

Note: All line numbers mentioned in the response letter refer to the line-numbered MARKED UP version of the manuscript.

This manuscript studies synchronization of the mechanical oscillation of an optomechanical system to a radiation pressure drive. Synchronization is an important and rich phenomenon, observed in many different contexts and with significant applications.

The paper is solid, including some very clean and elegant result together with a careful comparison of simulations to experiments. I find the synchronization data in Fig. 2 d2-g2) particularly beautiful. I commend the authors on a very nice set of experiments.

R: We appreciate the referee's enthusiasm regarding the quality of our experimental results.

R2-Q1) This said, I'm not sure that the work is suitable for Nature Communications. My most major concern is the level of advance reported in the manuscript. The physical technology (a silicon nitride double disk resonator) is well established, and the key results - that synchronization is possible by driving at harmonic and fractional frequencies - are extensions of prior work, rather than entirely new demonstrations. In particular, as acknowledged by the authors their Ref. [57] shows synchronization to a second harmonic drive. Less explicitly acknowledged by the authors is that their Ref. [58] shows synchronization to a fractional drive. I would note that Physical Review E 91 032910 (2015) also demonstrates second harmonic synchronization.

The present work goes further than these previous works, demonstrating synchronization up to the fourth harmonic, and showing explicitly through data processing that it is possible to do frequency division using harmonic synchronization. It also includes the interesting

observation that the synchronization bandwidth is larger for the second and fourth harmonic than for the fundamental, and that this has implication for the phase noise of the synchronized oscillator. Nevertheless, to me these all feel like solid progress, but not the dramatic leap one might expect for Nature Communications.

R: We agree with the reviewer that previous publications offered a glimpse into the potential of optomechanical systems to exhibit non-trivial synchronization response. However, we emphasize that previous demonstrations did not show evidence that a strong nontrivial synchronization in purely optomechanical systems was possible. For instance, in Ref.[57] no data, or theory is presented concerning the strength of the observed synchronization at half and twice the oscillator frequencies. Meanwhile, the two inspiring papers from Eyal Buks group (we have now also included Physical Review E 91 032910 (2015) - new Ref. [60]) rely on the low frequency thermal response of the mechanical resonator to achieve non-trivial synchronization. We modified the introduction around **lines 45-60** to give a more precise perspective on these works.

Our demonstration, based purely on the optical and mechanical responses of the system, is not only a semantic improvement over these previous works; instead, it is a result of the strong nonlinearity provided directly by radiation pressure (instead of depending on thermal response), ensuring enough bandwidth to respond to the higher harmonic injection signals. Therefore, it provides a direct route to scale these effects to even higher radio- frequencies. Since the effects reported here are not based on the heating of the mechanical mode (*S. De Liberato, N. Lambert, and F. Nori, "Quantum noise in photothermal cooling," Phys. Rev. A - At. Mol. Opt. Phys., vol. 83, no. 3, p. 033809, Mar. 2011.*), they are also more likely to be accessible in the quantum regime of optomechanical oscillators, experimentally enabling the field of nonlinear quantum synchronization.

R2-Q2) In my view, the most novel aspect of the publication is the explicit demonstration of frequency division (even if only through data processing). However, here I feel that the paper is lacking the rigor that is found elsewhere in it.

R: In order to build upon this positive perspective regarding our novel demonstration of frequency division we performed new experiments to strengthen our demonstration and, hopefully, address the referee's concern about the impact of this work.

In the new experiments we used an actual low-pass RF filter to capture the frequency divider signal for 2:1, 3:1, and 4:1 frequency division, experimentally demonstrating for the first time a purely optomechanical RF frequency divider.

R2-Q3) Perhaps the most important unanswered question is the relative quality (in terms of phase noise) of the divided signal compared to the input radio frequency signal. I would expect that this signal will be of far inferior quality, raising the question of the true usefulness of this form of optomechanical frequency division. Were the authors to do a direct comparison and show rigorously that the divided phase noise was comparable to that on the input radio frequency signal, or show through simulations/theory that there was a believable path to get there, this would significantly increase the novelty of the paper in my mind.

R: To address the phase-noise performance of our optomechanical divider, we present *new experimental data* in the revised Fig. 4, showing that improved phase noise performance is

possible even at the 4th harmonic by increasing the RF power by only 3 dB. In the revised Figure 4(a), we selected phase noise traces corresponding to a 3 dB higher RF power (-7 dBm, compared to -10 dBm in the original submission). While still in the weak-driving regime (5.5% of modulation depth), this injection strength is enough to induce the low-frequency phase-noise plateau for the 4th harmonic excitation. Further improvement below -100 dBc/Hz is achieved at higher RF powers, as shown in Fig. 4(b) and discussed around **lines 420-435**.

In the new supplementary information Sec. III -- dedicated to phase-noise -- we also show new data highlighting the reduction of phase-noise, as RF power is increased, at all observed harmonics.

Finally, we respectfully disagree with the referee regarding the usage of a well established optomechanical device. In our humble opinion, this further highlights that the higher order synchronization reported here could be easily translated to other optomechanical devices. Furthermore, we point out that dual-disk optomechanical resonators with mechanical quality factors over 10,000 (~10x larger than the current device) have been previously reported by us (Ref.[72]: M. Zhang, et al, *Appl. Phys. Lett.*, vol. 105, no. 5, 2014.). Such higher mechanical Q, associated with larger mechanical amplitudes (driven by stronger optical power), could result in 30 dB enhancement in the demonstrated phase-noise performance, as we analyze in the new Suppl. Information section III.

Some other points:

R2-Q4) Paragraph 3 of the introduction is not very clear. The first sentence says that higher-order synchronization has been "overlooked", Then "For instance..." lists examples where it has not been overlooked, rather than evidence that it has been overlooked as implied by the "For instance". As discussed above, there is also at least one reference missing here, as well an acknowledgement that Ref. [58] showed fractional synchronization.
Resp 4): We agree with the referee that the wording chosen in paragraph 3 was not adequate. We have modified this paragraph to include the suggested changes and more explicitly acknowledge Ref [58] (now Ref. 59) and the included Ref. 60.

R2-Q5) It isn't exactly clear to me what the vertical axis label "AT Tip (dBm)" means in Fig. 3d. I would define this explicitly in the caption.

Resp Q5): AT Tip (dBm) is the RF power associated with the appearance of the Arnold tongue. We have adjusted Fig.3d and associated captions to clarify this.

R2-Q6) The 4:1 phase noise curve in Fig. 4a seems qualitatively different than the other curves. Can the authors comment on this?

Resp Q6): The behavior observed for the 4:1 is actually present in all harmonics, at a sufficiently low RF power. Indeed, the lack of the plateau in the 4:1 phase-noise was merely due to the choice of a too low RF power. We chose now a 3 dB stronger RF tone where all PN traces exhibit the slow-frequency plateaus (similar to the 3:1 and 2:1 case). We have also included new data in Fig. 4(b) and Supplementary sec. III to demonstrate that transition to the low frequency plateau also occurs at other harmonics. This also highlights the

possibility of reducing phase-noise for any harmonic, at the expense of increasing the RF power.

The text was modified around **lines 325-330** to address this concern.

R2-Q7) The last paragraph on Page 6 comparing optomechanical and other frequency dividers would benefit from a clearer discussion of pros and cons. E.g. Why do OMO type dividers potentially offer lower power-consumption? ...and the authors point out a limitation of OMOs (I think, the wording isn't clear), that they have limited locking range. They then don't discuss how their approach, presumably, helps overcome this?

Resp Q7): We revised the discussion around page 6 to make a clearer statement about the divider performance. The initial statement on this paragraph was to highlight advantages of analog injection locking dividers compared to other technologies. We improved this paragraph and also added the following sentences (**lines 425-450**):

" Further improvement in phase-noise could be achieved by using devices with higher mechanical quality factor and stronger optical driving power, for instance, double-disk optomechanical devices with mechanical quality factors exceeding 10^4 and driven at larger amplitudes (using higher optical power) could exhibit a further PN reduction of 30dB (see Supplementary Information (Part III)). These results show that OMO-based frequency dividers can be readily derived from the observed higher-order synchronization. Although there is room for improvement in optomechanical frequency dividers, its ability to generate frequency references in the optical domain could be explored in experiments requiring optical synchronization, such as radio antenna telescopes [82], optical frequency combs [50], or coherently linking arrays of optomechanical oscillators with distinct frequencies [67]. Given the current state-of-the-art in hybrid integration [83] and electro-optical conversion in photonic circuits [73], the demonstrated divider could still ensure the low power consumption expected for injection locking frequency division."

REVIEWERS' COMMENTS

Reviewer #1 (Remarks to the Author):

The authors have replied to all the points I raised previously.
In my opinion, the manuscript is now suitable for publication in Nature Comm.

Reviewer #2 (Remarks to the Author):

The authors have put significant efforts into addressing the comments of the reviewers, and particularly into addressing the two primary concerns I had. With the new experimental data and analysis included in the paper, I am happy to recommend publication in Nature Communications.

We thank all referees for their voluntary careful reading and insightful feedback that helped improve our manuscript.